# Density Estimation via Discrepancy Based Adaptive Sequential Partition

**Dangna Li**
ICME,
Stanford University
Stanford, CA 94305
`dangna@stanford.edu`

**Kun Yang**
Google
Mountain View, CA 94043
`kunyang@stanford.edu`

**Wing Hung Wong**
Department of Statistics
Stanford University
Stanford, CA 94305
`whwong@stanford.edu`

## Abstract

Given $iid$ observations from an unknown absolute continuous distribution defined on some domain $\Omega$, we propose a nonparametric method to learn a piecewise constant function to approximate the underlying probability density function. Our density estimate is a piecewise constant function defined on a binary partition of $\Omega$. The key ingredient of the algorithm is to use discrepancy, a concept originates from Quasi Monte Carlo analysis, to control the partition process. The resulting algorithm is simple, efficient, and has a provable convergence rate. We empirically demonstrate its efficiency as a density estimation method. We also show how it can be utilized to find good initializations for k-means.

## 1 Introduction

Density estimation is one of the fundamental problems in statistics. Once an explicit estimate of the density function is constructed, various kinds of statistical inference tasks follow naturally. Given $iid$ observations, our goal in this paper is to construct an estimate of their common density function via a nonparametric domain partition approach.

As pointed out in [1], for density estimation, the bias due to the limited approximation power of a parametric family will become dominant in the over all error as the sample size grows. Hence it is necessary to adopt a nonparametric approach to handle this bias. The kernel density estimation [2] is a popular nonparametric density estimation method. Although in theory it can achieve optimal convergence rate when the kernel and the bandwidth are appropriately chosen, its result can be sensitive to the choice of bandwidth, especially in high dimension. In practice, kernel density estimation is typically not applicable to problems of dimension higher than 6.

Another widely used nonparametric density estimation method in low dimension is the histogram. But similarly with kernel density estimation, it can not be scaled easily to higher dimensions. Motivated by the usefulness of histogram and the need for a method to handle higher dimensional cases, we propose a novel nonparametric density estimation method which learns a piecewise constant density function defined on a binary partition of domain $\Omega$.

A key ingredient for any partition based method is the decision for stopping. Based on the observation that for any piecewise constant density, the distribution conditioned on each sub-region is uniform, we propose to use star discrepancy, which originates from analysis of Quasi-Monte Carlo methods, to formally measure the degree of uniformity. We will see in section 4 that this allows our density estimator to have near optimal convergence rate.

In summary, we highlight our contribution as follows:

- To the best of our knowledge, our method is the first density estimation method that utilizes Quasi-Monte Carlo technique in density estimation.

- We provide an error analysis on binary partition based density estimation method. We establish an $O(n^{-\frac{1}{2}})$ error bound for the density estimator. The result is optimal in the sense that essentially all Monte Carlo methods have the same convergence rate. Our simulation results support the tightness of this bound.
- One of the advantage of our method over existing ones is its efficiency. We demonstrate in section 5 that our method has comparable accuracy with other methods in terms of Hellinger distance while achieving an approximately $10^2$-fold speed up.
- Our method is a general data exploration tool and is readily applicable to many important learning tasks. Specifically, we demonstrate in section 5.3 how it can be used to find good initializations for k-means.

## 2  Related work

Existing domain partition based density estimators can be divided into two categories: the first category belongs to the Bayesian nonparametric framework. Optional Pólya Tree (OPT) [3] is a class of nonparametric conjugate priors on the set of piecewise constant density functions defined on some partition of $\Omega$. Bayesian Sequential Partitioning (BSP) [1] is introduced as a computationally more attractive alternative to OPT. Inferences for both methods are performed by sampling from the posterior distribution of density functions. Our improvement over these two methods is two-fold. First, we no longer restrict the binary partition to be always at the middle. By introducing a new statistic called the "gap", we allow the partitions to be adaptive to the data. Second, our method does not stem from a Bayesian origin and proceeds in a top down, greedy fashion. This makes our method computationally much more attractive than OPT and BSP, whose inference can be quite computationally intensive.

The second category is tree based density estimators [4] [5]. As an example, Density Estimation Trees [5] is generalization of classification trees and regression trees for the task of density estimation. Its tree based origin has led to a loss minimization perspective: the learning of the tree is done by minimizing the integrated squared error. However, the true loss function can only be approximated by a surrogate and the optimization problem is difficult to solve. The objective of our method is much simpler and leads to an intuitive and efficient algorithm.

## 3  Main algorithm

### 3.1  Notations and definitions

In this paper we consider the problem of estimating a joint density function $f$ from a given set of observations. Without loss of generality, we assume the data domain $\Omega = [0, 1]^d$, a hyper-rectangle in $\mathbb{R}^d$. We use the short hand notation $[a, b] = \prod_{j=1}^{d}[a_j, b_j]$ to denote a hyper-rectangle in $\mathbb{R}^d$, where $a = (a_1, \cdots, a_d), b = (b_1, \cdots, b_d) \in [0, 1]^d$. Each $(a_j, b_j)$ pair specifies the lower and upper bound of the hyper-rectangle along dimension $j$.

We restrict our attention to the class of piecewise constant functions after balancing the trade-off between simplicity and representational power: Ideally, we would like the function class to have concise representation while at the same time allowing for efficient evaluation. On the other hand, we would like to be able to approximate any continuous density function arbitrarily well (at least as the sample size goes to infinity). This trade-off has led us to choose the set of piecewise constant functions supported on binary partitions: First, we only need $2d + 1$ floating point numbers to uniquely define a sub-rectangle ($2d$ for its location and $1$ for its density value). Second, it is well known that the set of positive, integrable, piesewise constant functions is dense in $L^p$ for $p \in [1, \infty)$.

The binary partition we consider can be defined in the following recursive way: starting with $\mathcal{P}_0 = \Omega$. Suppose we have a binary partition $\mathcal{P}_t = \{\Omega^{(1)}, \cdots, \Omega^{(t)}\}$ at level $t$, where $\cup_{i=1}^{t}\Omega^{(i)} = \Omega$, $\Omega^{(i)} \cap \Omega^{(j)} = \varnothing$, $i \neq j$, a level $t + 1$ partition $\mathcal{P}_{t+1}$ is obtained by dividing one sub-rectangle $\Omega^{(i)}$ in $\mathcal{P}_t$ along one of its coordinates, parallel to one of the dimension. See Figure 1 for an illustration.

### 3.2  Adaptive partition and discrepancy control

The above recursive build up has two key steps. The first is to decide whether to further split a sub-rectangle. One helpful intuition is that for piecewise constant densities, the distribution conditioned on each sub-rectangle is uniform. Therefore the partition should stop when the points inside a sub-rectangle are approximatly uniformly scattered. In other words, we stop the partition when further

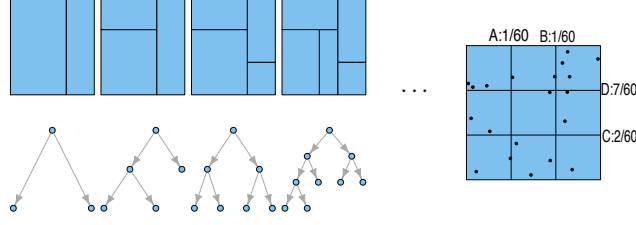

**Figure 1:** Left: a sequence of binary partition and the corresponding tree representation; if we encode partitioning information (e.g., the location where the split occurs) in the nodes, there is a one to one mapping between the tree representations and the partitions. Right: the gaps with $m = 3$, we split the rectangle at location D, which corresponds to the largest gap (Assuming it does not satisfy (2), see the text for more details)

.

partitioning does not reveal much additional information about the underlying density landscape. We propose to use star discrepancy, which is a concept originates from the analysis of Quasi-Monte Carlo methods, to formally measure the degree of uniformity of points in a sub-rectangle. Star discrepancy is defined as:

**Definition 1.** *Given $n$ points $X_n = \{x_1, ..., x_n\}$ in $[0, 1]^d$. The star discrepancy $D^*(X_n)$ is defined as:*

$$D^*(X_n) = \sup_{a \in [0,1]^d} \left| \frac{1}{n} \sum_{i=1}^{n} \mathbf{1}\{x_i \in [0, a)\} - \prod_{j=1}^{d} a_j \right| \tag{1}$$

The supremum is taken over all $d$-dimensional sub-rectangles $[0, a)$. Given star discrepancy $D^*(X_n)$, we have the following error bound for Monte Carlo integration (See [6] for a proof):

**Theorem 2.** *(Koksma-Hlawka inequality) Let $X_n = \{x_1, x_2, ..., x_n\}$ be a set of points in $[0, 1]^d$ with discrepancy $D^*(X_n)$; Let $f$ be a function on $[0, 1]^d$ of bounded variation $\mathcal{V}(f)$. Then,*

$$\left| \int_{[0,1]^d} f(x)dx - \frac{1}{n} \sum_{i=1}^{n} f(x_i) \right| \leq \mathcal{V}(f) D^*(X_n)$$

*where $\mathcal{V}(f)$ is the total variation in the sense of Hardy and Krause (See [7] for its precise definition).*

The above theorem implies if the star discrepancy $D^*(X_n)$ is under control, the empirical distribution will be a good approximation to the true distribution. Therefore, we may decide to keep partitioning a sub-rectangle until its discrepancy is lower than some threshold. We shall see in section 4 that this provably guarantees our density estimate is a good approximation to the true density function.

Another important ingredient of all partition based methods is the choice of splitting point. In order to find a good location to split for $[a, b] = \prod_{j=1}^{d}[a_j, b_j]$, we divide $j^{th}$ dimension into $m$ equal-sized bins: $[a_j, a_j + (b_j - a_j)/m], ..., [a_j + (b_j - a_j)(m-2)/m, a_j + (b_j - a_j)(m-1)/m]$ and keep track of the gaps at $a_j + (b_j - a_j)/m, ..., a_j + (b_j - a_j)(m-1)/m$, where the gap $g_{jk}$ is defined as $|(1/n) \sum_{i=1}^{n} \mathbf{1}(x_{ij} < a_j + (b_j - a_j)k/m) - k/m|$ for $k = 1, ..., (m-1)$, there are total $(m-1)d$ gaps recorded (Figure 1). Here $m$ is a hyper-parameter chosen by the user. $[a, b]$ is split into two sub-rectangles along the dimension and location corresponding to maximum gap (Figure 1). The pseudocode for the complete algorithm is given in Algorithm 1. We refer to this algorithm as DSP in the sequel. One distinct feature of DSP is it only requires the user to specify two parameters: $m, \theta$, where $m$ is the number of bins along each dimension; $\theta$ is the parameter for discrepancy control (See theorem 2 for more details). In some applications, the user may prefer putting an upper bound on the number of total partitions. In that case, there is typically no need to specify $\theta$. Choices for these parameters are discussed in Section 5.

The resulting density estimates $\hat{p}$ is a piecewise constant function defined on a binary partition of $\Omega$: $\hat{p}(x) = \sum_{i=1}^{L} d(r_i)\mathbf{1}\{x \in r_i\}$ where $\mathbf{1}$ is the indicator function; $L$ is the total number of sub-rectangles in the final partition; $\{r_i, d(r_i)\}_{i=1}^{L}$ are the sub-rectangle and density pairs. We demonstrate in section 5 how $\hat{p}(x)$ can be leveraged to find good initializations for k-means. In the following section, we establish a convergence result of our density estimator.

---

**Algorithm 1** Density Estimation via Discrepancy Based Sequential Partition (DSP)

---

**Input**: $X_N, m, \theta$
**Output**: A piecewise constant function $\Pr(\cdot)$ defined on a binary partition $\mathcal{R}$
Let $\Pr(r)$ denote the probability mass of region $r \subset \Omega$; let $X_N(r)$ denote the points in $X_N$ that lie within $r$, where $r \subset \Omega$. $n_i$ denotes the size of set $X^{(i)}$.

1: **procedure** DSP($X_N, m, \theta$)
2:     $\mathcal{B} = \{[0,1]^d\}, \Pr([0,1]^d) = 1$
3:     **while** true **do**
4:         $\mathcal{R}' = \varnothing$
5:         **for** each $r_i = [a^{(i)}, b^{(i)}]$ in $\mathcal{R}$ **do**
6:             Calculate gaps $\{g_{jk}\}_{j=1,...,d, k=1,...,m-1}$
7:             Scale $X(r_i) = \{x_{i_l}\}_{l=1}^{n_i}$ to $\tilde{X}^{(i)} = \{\tilde{x}_{i_l} = (\frac{x_{i_l,1}-a_1^{(i)}}{b_1^{(i)}}, ..., \frac{x_{i_l,d}-a_d^{(i)}}{b_d^{(i)}})\}_{l=1}^{n_i}$
8:             **if** $X(r_i) \neq \varnothing$ and $D^*(\tilde{X}^{(i)}) > \theta\sqrt{N}/n_i$ **then**         ▷ Condition (2) in Theorem 4
9:                 Split $r_i$ into $r_{i_1} = [a^{(i_1)}, b^{(i_1)}]$ and $r_{i_2} = [a^{(i_2)}, b^{(i_2)}]$ along the max gap (Figure 1).
10:                 $\Pr(r_{i_1}) = \Pr(r_i)\frac{|P(r_{i_1})|}{n_i}, \Pr(r_{i_2}) = \Pr(r_i) - \Pr(r_{i_1})$
11:                 $\mathcal{R}' = \mathcal{R}' \cup \{r_{i_1}, r_{i_2}\}$
12:             **else** $\mathcal{R}' = \mathcal{R}' \cup \{r_i\}$
13:         **if** $\mathcal{R}' \neq \mathcal{R}$ **then** $\mathcal{R} = \mathcal{R}'$
14:         **else** return $\mathcal{R}, \Pr(\cdot)$

---

# 4 Theoretical results

Before we establish our main theorem, we need the following lemma:[1]

**Lemma 3.** *Let $D_n^* = \inf_{\{x_1,...,x_n\} \in [0,1]^d} D^*(x_1, ..., x_n)$, then we have*

$$D_n^* \leq c\sqrt{\frac{d}{n}}$$

*for all $n, d \in \mathbb{R}^+$, where $c$ is some positive constant.*

We now state our main theorem:

**Theorem 4.** *Let $f$ be a function defined on $\Omega = [0,1]^d$ with bounded variation. Let $X_N = \{x_1, ..., x_N \in \Omega\}$ and $\{[a^{(i)}, b^{(i)}], i = 1, \cdots, L\}$ be a level $L$ binary partition of $\Omega$. Further denote by $X^{(i)} = \{x_j = (x_{j1}, ..., x_{jd}), x_j \in [a^{(i)}, b^{(i)}]$ and $\} \cap X_N$, i.e. the part of $X_N$ in sub-rectangle $i$. $n_i = |X^{(i)}|$. Suppose in each sub-rectangle $[a^{(i)}, b^{(i)}]$, $X^{(i)}$ satisfies*

$$D^*(\tilde{X}^{(i)}) \leq \alpha^{(i)} D_{n_i}^* \tag{2}$$

*where $\tilde{X}^{(i)} = \{\tilde{x}_j = (\frac{x_{j1}-a_1^{(i)}}{b_1^{(i)}}, ..., \frac{x_{jd}-a_d^{(i)}}{b_d^{(i)}}), x_j \in X^{(i)}\}$, $\alpha^{(i)} = \sqrt{\frac{N}{n_i d}}\frac{\theta}{c}$ for some positive constant $\theta$, $D_{n_i}^*$ is defined as in lemma 3. Then*

$$\left| \int_{[0,1]^d} f(x)\hat{p}(x)dx - \frac{1}{N}\sum_{i=1}^{N} f(x_i) \right| \leq \frac{\theta}{\sqrt{N}}\mathcal{V}(f) \tag{3}$$

*where $\hat{p}(x)$ is a piecewise constant density estimator given by*

$$\hat{p}(x) = \sum_{i=1}^{L} d_i \mathbf{1}\{x \in [a^{(i)}, b^{(i)}]\}$$

*with $d_i = (\prod_{j=1}^{d}(b_j^{(i)} - a_j^{(i)}))^{-1} n_i/N$, i.e., the empirical density.*

In the above theorem, $\alpha^{(i)}$ controls the relative uniformity of the points and is adaptive to $X^{(i)}$. It imposes more restrictive constraints on regions containing larget proportion of the sample ($n_i/N$). Although our density estimate is not the only estimator which satisfies (3), (for example, both the empirical distribution in the asymptotic limit and kernel density estimator with sufficiently small bandwidth meet the criterion), one advantage of our density estimator is that it provides a very concise

summary of the data while at the same time capturing the landscape of the underlying distribution. In addition, the piecewise constant function does not suffer from having too many "local bumps", which is a common problem for kernel density estimator. Moreover, under certain regularity conditions (e.g. bounded second moments), the convergence rate of Monte Carlo methods for $\frac{1}{N}\sum_{i=1}^{N}f(x_i)$ to $\int_{[0,1]^d}f(x)p(x)dx$ is of order $O(N^{-\frac{1}{2}})$. Our density estimate is optimal in the sense that it achieves the same rate of convergence. Given theorem 4, we have the following convergence result:

**Corollary 5.** *Let $\hat{p}(x)$ be the estimated density function as in theorem 4. For any hyper-rectangle $A = [a, b] \subset [0, 1]^d$, let $\hat{P}(A) = \int_A \hat{p}(x)dx$ and $P(A) = \int_A p(x)dx$, then*

$$\sup_{A \subset [0,1]^d} |\hat{P}(A) - P(A)| \to 0$$

*at the order $O(n^{-\frac{1}{2}})$.*

**Remark 4.1.** *It is worth pointing out that the total variation distance between two probability measures $\hat{P}$ and $P$ is defined as $\delta(\hat{P}, P) = \sup_{A \in \mathcal{B}} |\hat{P}(A) - P(A)|$, where $\mathcal{B}$ is the Borel $\sigma$-algebra of $[0, 1]^d$. In contrast, Corollary 5 restricts $A$ to be hyper-rectangles.*

## 5 Experimental results

### 5.1 Implementation details

In some applications, we find it helpful to first estimate the marginal densities for each component variables $x_{\cdot j}$ ($j = 1, ..., d$), then make a copula transformation $z_{\cdot j} = \hat{F}_j(x_{\cdot j})$, where $\hat{F}_j$ is the estimated cdf of $x_{\cdot j}$. After such a transformation, we can take the domain to be $[0, 1]^d$. Also we find this can save the number of partition needed by DSP. Unless otherwise stated, we use copula transform in our experiments whenever the dimension exceeds 3.

We make the following observations to improve the efficiency of DSP: 1) First observe that $\max_{j=1,...,d} D^*(\{x_{ij}\}_{i=1}^n) \leq D^*(\{x_i\}_{i=1}^n)$. Let $x_{(i)j}$ be the $i$th smallest element in $\{x_{ij}\}_{i=1}^n$, then $D^*(\{x_{ij}\}_{i=1}^n) = \frac{1}{2n} + \max_i |x_{(i)j} - \frac{2i-1}{2n}|$ [9], which has complexity $O(n \log n)$. Hence $\max_{j=1,...,d} D^*(\{x_{ij}\}_{i=1}^n)$ can be used to compare against $\theta\sqrt{L}/n$ first before calculating $D^*(\{x_i\}_{i=1}^n)$; 2) $\theta\sqrt{N}/n$ is large when $n$ is small, but $D^*(\{x_i\}_{i=1}^n)$ is bounded above by 1; 3) $\theta\sqrt{N}/n$ is tiny when $n$ is large and $D^*(\{x_i\}_{i=1}^n)$ is bounded below by $c_d \log^{(d-1)/2} n\,n^{-1}$ with some constant $c_d$ depending on $d$ [10]; thus we can keep splitting without checking (2) when $\theta\sqrt{N}/n \leq \epsilon$, where $\epsilon$ is a small positive constant (say 0.001) specified by the user. This strategy has proved to be effective in decreasing the runtime significantly at the cost of introducing a few more sub-rectangles.

Another approximation works well in practice is by replacing star discrepancy with computationally attractive $\mathcal{L}_2$ star discrepancy, i.e., $D^{(2)}(X_n) = (\int_{[0,1]^d} |\frac{1}{n}\sum_{i=1}^n \mathbf{1}_{x_i \in [0,a)} - \prod_{i=1}^d a_i|^2 da)^{\frac{1}{2}}$; in fact, several statistics to test uniformity hypothesis based on $D^{(2)}$ are proposed in [11]; however, the theoretical guarantee in Theorem 4 no longer holds. By Warnock's formula [9],

$$[D^{(2)}(X_n)]^2 = \frac{1}{3^d} - \frac{2^{1-d}}{n}\sum_{i=1}^n\prod_{j=1}^d(1 - x_{ij}^2) + \frac{1}{n^2}\sum_{i,l=1}^n\prod_{j=1}^d \min\{1 - x_{ij}, 1 - x_{lj}\}$$

$D^{(2)}$ can be computed in $O(n \log^{d-1} n)$ by K. Frank and S. Heinrich's algorithm [9]. At each scan of $\mathcal{R}$ in Algorithm 1, the total complexity is at most $\sum_{i=1}^L O(n_i \log^{d-1} n_i) \leq \sum_{i=1}^L O(n_i \log^{d-1} N) \leq O(N \log^{d-1} N)$.

There are no closed form formulas for calculating $D^*(X_n)$ and $D_n^*$ except for low dimensions. If we replace $\alpha^{(i)}$ in (2) and apply Lemma 3, what we are actually trying to do is to control $D^*(\tilde{X}^{(i)})$ by $\theta\sqrt{N}/n_i$. There are many existing work on ways to approximate $D^*(X_n)$. In particular, a new randomized algorithm based on threshold accepting is developed in [12]. Comprehensive numerical tests indicate that it improves upon other algorithms, especially in when $20 \leq d \leq 50$. We used this algorithm in our experiments. The interested readers are referred to the original paper for more details.

## 5.2 DSP as a density estimate

**1)** To demonstrate the method and visualize the results, we apply it on several 2-dimensional data sets simulated from 3 distributions with different geometry:

1. Gaussian: $x \sim \mathcal{N}(\mu, \Sigma)\mathbf{1}\{x \in [0,1]^2\}$, with $\mu = (.5, .5)^T$, $\Sigma = [0.08, 0.02; 0.02, 0.02]$
2. Mixture of Gaussians: $x \sim \frac{1}{2}\sum_{i=1}^{2}\mathcal{N}(\mu_i, \Sigma_i)\mathbf{1}\{x \in [0,1]^2\}$ with $\mu_1 = (.50, .25)^T$, and $\mu_2 = (.50, .75)^T, \Sigma_1 = \Sigma_2 = [0.04, 0.01; 0.01, 0.01]$;
3. Mixture of Betas: $x \sim \frac{1}{3}(\text{beta}(2,5)\text{beta}(5,2) + \text{beta}(4,2)\text{beta}(2,4) + \text{beta}(1,3)\text{beta}(3,1))$;

where $\mathcal{N}(\mu, \Sigma)$ denotes multivariate Gaussian distribution and $\text{beta}(\alpha, \beta)$ denotes beta distribution. We simulated $10^5$ points for each distribution. See the first row of Figure 2 for visualizations of the estimated densities. The figure shows DSP accurately estimates the true density landscape in these three toy examples.

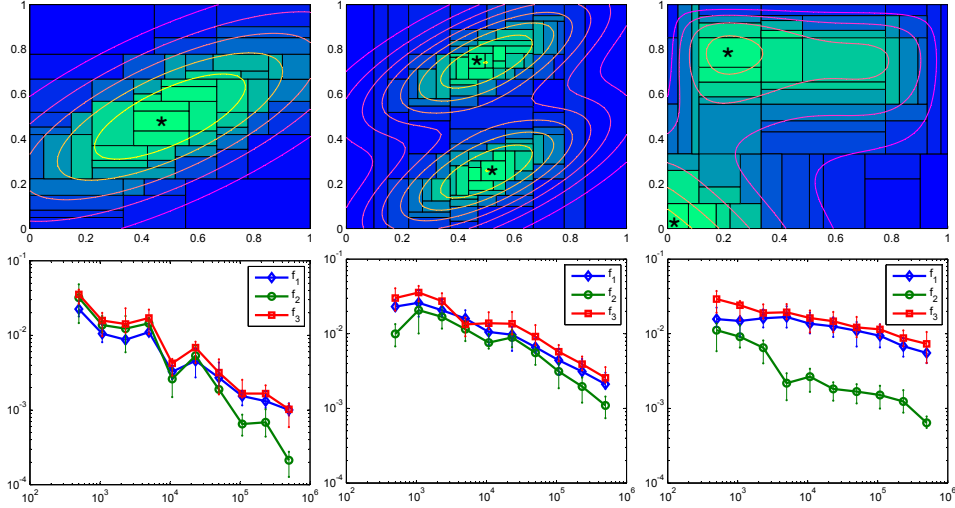

**Figure 2: First row**: estimated densities for 3 simulated 2D datasets. The modes are marked with stars. The corresponding contours of true densities are embedded for comparison. **Second row**: simulation of 2, 5 and 10 dimensional cases (from left to right) with reference functions $f_1, f_2, f_3$. $x$-axis: sample size $n$. $y$-axis: error between the true integral and the estimated integral. The vertical bars are standard error bars obtained from 10 replications. See section 5.2 2) for more details.

**2)** To evaluate the theoretical bound (3), we choose the following three 3 reference functions with dimension $d = 2, 5$ and $10$ respectively: $f_1(x) = \sum_{i=1}^{n}\sum_{j=1}^{d}x_{ij}^{\frac{1}{2}}$, $f_2(x) = \sum_{i=1}^{n}\sum_{j=1}^{d}x_{ij}$, $f_3(x) = (\sum_{i=1}^{n}\sum_{j=1}^{d}x_{ij}^{\frac{1}{2}})^2$. We generate $n \in \{10^2, 10^3, 10^4, 10^5, 10^6\}$ samples from $p(x) = \frac{1}{2}\Big(\prod_{j=1}^{d}\text{beta}(x_j, 15, 5) + \prod_{j=1}^{d}\text{beta}(x_j, 5, 15)\Big)$, where $\text{beta}(\cdot, \alpha, \beta)$ is the density function of beta distribution.

The error $|\int_{[0,1]^d}f_k(x)p(x)dx - \int_{[0,1]^d}f_k(x)\hat{p}(x)dx|$ is bounded by $|\int_{[0,1]^d}f_k(x)p(x)dx - \frac{1}{n}\sum_{j=1}^{n}f_k(x_j)| + |\int_{[0,1]^d}f_k(x)\hat{p}(x)dx - \frac{1}{n}\sum_{j=1}^{n}f_k(x_j)|$ where $\hat{p}(x)$ is the estimated density; For almost all Monte Carlo methods, the first term is of order $O(n^{-\frac{1}{2}})$. The second term is controlled by (3). Thus in total the error is of order $O(n^{-\frac{1}{2}})$. We have plot the error against the sample size on log-log scale for each dimension in the second row of Figure 2. The linear trends in the plots corroborate the bound in (3).

**3)** To show the efficiency and scalability of DSP, we compare it with KDE, OPT and BSP in terms of estimation error and running time. We simulate samples from $x \sim (\sum_{i=1}^{4}\pi_i\mathcal{N}(\mu_i, \Sigma_i))\mathbf{1}\{x \in [0,1]^d\}$ with $d = \{2, 3, \cdots, 6\}$ and $N = \{10^3, 10^4, 10^5\}$ respectively. The estimation error measured in terms of Hellinger Distance is summarized in Table 1. We set $m = 10, \theta = 0.01$ in our experiments. We found the resulting Hellinger distance to be quite robust as $m$ ranges from 3 to 20 (equally

spaced). The supplementary material includes the exact details about the parameters of the simulating distributions, estimation of Hellinger distance and other implementation details for the algorithms. The table shows DSP achieves comparable accuracy with the best of the other three methods. As mentioned at the beginning of this paper, one major advantage of DSP's is its speed. Table 2 shows our method achieves a significant speed up over all other three algorithms.

**Table 1: Error in Hellinger Distance** between the true density and KDE, OPT, BSP, our method for each $(d, n)$ pair. The numbers in parentheses are standard errors from 20 replicas. The best of the four method is highlighted in bold. Note that the simulations, being based on mixtures of Gaussians, is unfavorable for methods based on domain partitions.

| d | Hellinger Distance ($n = 10^3$) | | | | Hellinger Distance ($n = 10^4$) | | | | Hellinger Distance ($n = 10^5$) | | | |
|---|---|---|---|---|---|---|---|---|---|---|---|---|
|   | KDE | OPT | BSP | DSP | KDE | OPT | BSP | DSP | KDE | OPT | BSP | DSP |
| 2 | 0.2331 | **0.2147** | 0.2533 | 0.2634 | 0.1104 | 0.0957 | 0.1222 | **0.0803** | **0.0305** | 0.0376 | 0.0345 | 0.0312 |
|   | (0.0421) | (0.0172) | (0.0163) | (0.0207) | (0.0102) | (0.0036) | (0.0043) | (0.0013) | (0.0021) | (0.0021) | (0.0025) | (0.0027) |
| 3 | **0.2893** | 0.3279 | 0.2983 | 0.3072 | 0.2003 | 0.1722 | **0.1717** | 0.1721 | 0.1466 | 0.1117 | 0.1323 | **0.1020** |
|   | (0.0227) | (0.0128) | (0.0133) | (0.0265) | (0.0199) | (0.0028) | (0.0083) | (0.0073) | (0.0047) | (0.0008) | (0.0009) | (0.004) |
| 4 | 0.3913 | **0.3839** | 0.3872 | 0.3895 | **0.2466** | 0.2726 | 0.2882 | 0.2955 | 0.1900 | 0.1880 | 0.2100 | **0.1827** |
|   | (0.0325) | (0.0136) | (0.0117) | (0.0191) | (0.0113) | (0.0047) | (0.0065) | | (0.0057) | (0.0006) | (0.0006) | (0.0059) |
| 5 | 0.4522 | 0.4748 | 0.4435 | **0.4307** | 0.3599 | **0.3562** | 0.3987 | 0.3563 | **0.2817** | 0.2822 | 0.2916 | 0.2910 |
|   | (0.0317) | (0.009) | (0.0167) | (0.0302) | (0.0199) | (0.0025) | (0.0022) | (0.0031) | (0.0088) | (0.0005) | (0.0003) | (0.0002) |
| 6 | 0.5511 | **0.5508** | 0.5515 | 0.5527 | 0.4833 | 0.4015 | 0.4093 | **0.3911** | 0.3697 | **0.3409** | 0.3693 | 0.3701 |
|   | (0.0318) | (0.0307) | (0.0354) | (0.0381) | (0.0255) | (0.0023) | (0.0046) | (0.0037) | (0.0122) | (0.0005) | (0.0004) | (0.0002) |

**Table 2: Average CPU time in seconds** of KDE, OPT, BSP and our method for each $(d, n)$ pair. The numbers in parentheses are standard errors from 20 replicas. The speed-up is the fold speed-up computed as the ratio between the minimum run time of the other three methods and the run time of DSP. All methods are implemented in C++. See the supplementary material for more details.

| d | Running time ($n = 10^3$) | | | | | Running time ($n = 10^4$) | | | | | Running time ($n = 10^5$) | | | | |
|---|---|---|---|---|---|---|---|---|---|---|---|---|---|---|---|
|   | KDE | OPT | BSP | DSP | speed-up | KDE | OPT | BSP | DSP | speed-up | KDE | OPT | BSP | DSP | speed-up |
| 2 | 2.445 | 9.484 | 0.833 | 0.020 | 41 | 21.903 | 31.561 | 1.445 | 0.033 | 43 | 230.179 | 44.561 | 7.750 | 0.242 | 33 |
|   | (0.191) | (0.029) | (0.006) | (0.002) | | (1.905) | (0.079) | (0.014) | (0.002) | | (130.572) | (0.639) | (0.178) | (0.015) | |
| 3 | 2.655 | 25.073 | 1.054 | 0.019 | 55 | 26.964 | 36.683 | 2.819 | 0.044 | 64 | 278.075 | 56.329 | 21.104 | 0.378 | 55 |
|   | (0.085) | (0.056) | (0.010) | (0.002) | | (1.089) | (0.076) | (0.036) | (0.001) | | (10.576) | (0.911) | (0.576) | (0.011) | |
| 4 | 3.540 | 32.112 | 1.314 | 0.019 | 69 | 37.141 | 39.219 | 5.861 | 0.049 | 119 | 347.501 | 67.366 | 53.620 | 0.485 | 108 |
|   | (0.116) | (0.072) | (0.014) | (0.002) | | (2.244) | (0.221) | (0.076) | (0.002) | | (14.676) | (3.018) | (2.917) | (0.018) | |
| 5 | 4.107 | 37.599 | 1.713 | 0.020 | 85 | 45.580 | 44.520 | 12.220 | 0.078 | 157 | 412.828 | 77.776 | 115.869 | 0.706 | 110 |
|   | (0.110) | (0.088) | (0.019) | (0.002) | | (2.124) | (0.587) | (0.154) | (0.002) | | (16.252) | (2.215) | (6.872) | (0.051) | |
| 6 | 4.986 | 41.565 | 2.749 | 0.020 | 137 | 53.291 | 43.032 | 21.696 | 0.127 | 170 | 519.298 | 81.023 | 218.999 | 0.896 | 90 |
|   | (0.214) | (0.147) | (0.024) | (0.001) | | (2.767) | (0.413) | (0.213) | (0.004) | | (29.276) | (3.703) | (6.046) | (0.071) | |

## 5.3 DSP-kmeans

In addition to being a competitive density estimator, we demonstrate in this section how DSP can be used to get good initializations for k-means. The resulting algorithm is referred to as DSP-kmeans.

Recall that given a fixed number of clusters $K$, the goal of k-means is to minimize the following objective function:

$$J_K \triangleq \sum_{k=1}^{K} \sum_{i \in C_k} \|x_i - m_k\|_2^2 \qquad (4)$$

where $C_k$ denote the set of points in cluster $k$; $\{m_k\}_{k=1}^{K}$ denote the cluster means. The original k-means algorithms proceeds by alternating between assigning points to centers and recomputing the means. As a result, the final clustering is usually only a local optima and can be sensitive to the initializations. Finding a good initialization has attracted a lot of attention over the past decade and now there is a descent number existing methods, each with their own perspectives. Below we review a few representative types.

One type of methods look for good initial centers sequentially. The idea is once the first center is picked, the second should be far away from the one that is already chosen. A similar argument applies to the rest of the centers. [13] [14] fall under this category. Several studies [15] [16] borrow ideas from hierarchical agglomerative clustering (HAC) to look for good initializations. In our experiments we used the algorithm described in [15]. One essential ingredient of this type of algorithms is the inter cluster distance, which could be problem dependent. Last but not least, there is a class of methods that attempt to utilize the relationship between PCA and k-means. [17] proposes a PCA-guided search for initial centers. [18] combines the relationship between PCA and k-means to look for good initialization. The general idea is to recursively splitting a cluster according the first principal component. We refer to this algorithm as PCA-REC.

DSP-kmeans is different from previous methods in that it tackles the initialization problem from a density estimation point of view. The idea behind DSP-kmeans is that cluster centers should be close to the modes of underlying probability density function. If a density estimator can accurately locate the modes of the underlying true density function, it should also be able to find good cluster centers. Due to its concise representation, DSP can be used for finding initializations for k-means in the following way: Suppose we are trying to cluster a dataset $Y$ with $K$ clusters. We first apply DSP on $Y$ to find a partition with $K$ non-empty sub-rectangles, i.e. sub-rectangles that have at least one point from $Y$. The output of DSP will be $K$ sub-rectangles. Denote the set of indices for the points in sub-rectangle $j$ by $S_j$, $j = 1, \ldots, K$, let $I_j = \frac{1}{|S_j|} \sum_{i \in S_j} Y_i$, i.e. $I_j$ is the sample average of points fall into sub-rectangle $j$. We then use $\{I_1, \cdots, I_K\}$ to initialize k-means. We also explored the following two-phase procedure: first over partition the space to build a more accurate density estimate. Points in different sub-rectangles are considered to be in different clusters. Then we merge the sub-rectangles hierarchically based on some measure of between cluster distance. We have found this to be helpful when the number of clusters $K$ is relatively small. For completeness, we have included the details of this two-phase DSP-kmeans in the supplementary material.

We test DSP-kmeans on 4 real world datasets of various number of data points and dimensions. Two of them are taken from the UCI machine learning repository [19]; the stem cell data set is taken from the FlowCAP challenges [20]; the mouse bone marrow data set is a recently published single-cell dataset measured using mass cytometry [21]. We use random initialization as the base case and compare it with DSP-kmeans, k-means++, PCA-REC and HAC. The numbers in Table 3 are the improvements in k-means objective function of a method over random initialization. The result shows when the number of clusters is relatively large DSP-kmeans achieves lower objective value in these four datasets. Although in theory almost all density estimator could be used to find good

**Table 3: Comparison of different initialization methods**. The number for method $j$ is relative to random initialization: $\frac{J_{K,j} - J_{K,0}}{J_{K,0}}$, where $J_{K,j}$ is the k-means objective value of method $j$ at convergence. Here we use 0 as index for random initialization. Negative number means the method perform worse than random initialization.

| | | Improvement over random init. | | | | | | Improvement over random init. | | | |
|---|---|---|---|---|---|---|---|---|---|---|---|
| Road network | k | k-means++ | PCA-REC | HAC | DSP-kmeans | Mouse bone marrow | k | k-means++ | PCA-REC | HAC | DSP-kmeans |
| n  4.3e+04 | 4 | 0.0 | -0.02 | **0.01** | 0.0 | n  8.7e+04 | 4 | **1.51** | 0.03 | 1.25 | 0.4 |
| d  3 | 10 | 0.0 | -0.12 | **0.25** | 0.08 | d  39 | 10 | 0.45 | 0.24 | 0.77 | **0.83** |
| | 20 | 0.43 | -0.46 | 1.68 | 2.04 | | 20 | 0.63 | -1.2 | 0.68 | **0.79** |
| | 40 | 11.7 | -2.52 | 2.27 | 13.62 | | 40 | 1.99 | -3.56 | 2.06 | **2.55** |
| | 60 | 19.78 | -3.45 | 18.69 | **20.91** | | 60 | 2.48 | -5.25 | 2.57 | **2.65** |
| Stem cell | k | k-means++ | PCA-REC | HAC | DSP-kmeans | US census | k | k-means++ | PCA-REC | HAC | DSP-kmeans |
| n  9.9e+03 | 4 | 3.45 | -2.1 | 3.67 | **3.96** | n  2.4e+06 | 4 | **47.44** | -2.33 | 46.72 | 40.44 |
| d  6 | 10 | **3.82** | -4.2 | 3.79 | 3.6 | d  68 | 10 | 40.52 | -1.9 | **41.48** | 39.52 |
| | 20 | **9.96** | -3.59 | 9.91 | 9.39 | | 20 | 32.63 | -1.97 | 29.49 | 32.55 |
| | 40 | 9.95 | -6.39 | 10.11 | **12.49** | | 40 | 32.66 | -5.15 | 33.41 | **34.61** |
| | 60 | 6.12 | -7.29 | 8.19 | **13.7** | | 60 | **21.7** | -1.19 | 16.28 | 21.68 |

initializations. Based on the comparison of Hellinger distance in Table 1, we would expect them to have similar performances. However, for OPT and BSP, their runtime would be a major bottleneck for their applicability The situation for KDE is slightly more complicated: not only it is computationally quite intensive, its output can not be represented as concisely as partition based methods. Here we see that the efficiency of DSP makes it possible to utilize it for other machine learning tasks.

## 6  Conclusion

In this paper we propose a novel density estimation method based on ideas from Quasi-Monte Carlo analysis. We prove it achieves a $O(n^{-\frac{1}{2}})$ error rate. By comparing it with other density estimation methods, we show DSP has comparable performance in terms of Hellinger distance while achieving a significant speed-up. We also show how DSP can be used to find good initializations for k-means. Due to space limitation, we were unable to include other interesting applications including mode seeking, data visualization via level set tree and data compression [22].

**Acknowledgements.** This work was supported by NIH-R01GM109836, NSF-DMS1330132 and NSF-DMS1407557. The second author's work was done when the author was a graduate student at Stanford University.

## Footnotes

[1]The proof for Lemma 3 can be found in [8]. Theorem 4 and Corollary 5 are proved in the supplementary material.

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
