[Supplementary Material]

# Supplementary Material for Density Estimation via Discrepancy Based Adaptive Sequential Partition

**Dangna Li**
ICME,
Stanford University
Stanford, CA 94305
dangna@stanford.edu

**Kun Yang**
Google
Mountain View, CA 94043
kunyang@stanford.edu

**Wing Hung Wong**
Department of Statistics
Stanford University
Stanford, CA 94305
whwong@stanford.edu

## 1 Proofs for Theorem 4 and Corollary 5

Before proving our main theorem, we need additionally the following three lemmas. The proofs for Lemma 1 and Lemma 2 are trivial and can be found in [1]. Lemma 3 is proved below.

**Lemma 1.** *Let $f$ be a function defined on a hyper-rectangle $[a, b] \in \mathbb{R}^d$. Let $\{[a^{(i)}, b^{(i)}] : 1 \leq i \leq L\}$ be a binary partition of $[a, b]$. Then*

$$\sum_{i=1}^{L} \mathcal{V}^{[a^{(i)}, b^{(i)}]}(f) = \mathcal{V}^{[a,b]}(f)$$

*where $\mathcal{V}^{[a,b]}$ denotes the total variation of $f$ with its domain restricting to $[a, b] \subset \Omega$.*

**Lemma 2.** *Let $f$ be a function defined on hyper-rectangle $[a, b] \in \mathbb{R}^d$. let $\tilde{f}$ be a function defined on hyper-rectangle $[\tilde{a}, \tilde{b}]$ with $\tilde{f}(x) \triangleq f \circ \phi(x)$, where each $\phi_i$ is a strictly monotone (increasing or decreasing) function from $[\tilde{a}^{(i)}, \tilde{b}^{(i)}]$ onto $[a^{(i)}, b^{(i)}]$, then*

$$\mathcal{V}^{[\tilde{a}, \tilde{b}]}(\tilde{f}) = \mathcal{V}^{[a,b]}(f)$$

**Lemma 3.** *Let $f$ be a function defined on a $d-$dimensional hyper-rectangle $[a, b]$. Let $X_n = \{x_1, ..., x_n \in [a, b]\}$. Then we have*

$$\Big| \int_{[a,b]} f(x)dx - \frac{\prod_{j=1}^{d}(b_j - a_j)}{n} \sum_{i=1}^{n} f(x_i) \Big| \leq \prod_{j=1}^{d}(b_j - a_j) D_n^*(\tilde{X}_n) \mathcal{V}^{[a,b]}(f) \tag{1}$$

*where $\tilde{X}_n = \{\tilde{x}_i = (\frac{x_{i1} - a_1}{b_1}, ..., \frac{x_{id} - a_d}{b_d}), x_i \in X_n\}_{i=1}^{n}$*

*Proof.* Define $\tilde{f}(\tilde{x}) = f(x)$, where $\tilde{x} = (\frac{x_1 - a_1}{b_1}, ..., \frac{x_d - a_d}{b_d})$. Then by Theorem 2, we have

$$\Big| \int_{[0,1]^d} \tilde{f}(\tilde{x})d\tilde{x} - \frac{1}{n} \sum_{i=1}^{n} \tilde{f}(\tilde{x}_i) \Big| \leq D_n^*(\tilde{X}_n) \mathcal{V}^{[0,1]^d}(\tilde{f})$$

By Lemma 2, we have:

$$\mathcal{V}^{[0,1]^d}(\tilde{f}) = \mathcal{V}^{[a,b]}(f)$$

By a change of variable, we have

$$\int_{[0,1]^d} \tilde{f}(\tilde{x})d\tilde{x} = \frac{1}{\prod_{j=1}^{d}(b_j - a_j)} \int_{[a,b]} f(x)dx$$

Notice $\tilde{f}(\tilde{x}_i) = f(x_i)$ by definition. Hence, (1) follows immediately. □

We now prove Theorem 4 and Corollary 5:

## 1.1 Proof for Theorem 4

*Proof.* Appling Theorem 3 to each $[a^{(i)}, b^{(i)}]$, $i = 1, ..., L$, we have

$$\Big| \int_{[a^{(i)},b^{(i)}]} f(x)dx - \frac{\prod_{j=1}^{d}(b_j^{(i)} - a_j^{(i)})}{n_i} \sum_{i=1}^{n_i} f(x_j^{(i)}) \Big| \leq \prod_{j=1}^{d}(b_j^{(i)} - a_j^{(i)})D_{n_i}^*(\tilde{X}^{(i)})\mathcal{V}^{[a^{(i)},b^{(i)}]}(f) \quad (2)$$

By triangular inequality, we have

$$\Big| \int_{[0,1]^d} f(x)\hat{p}(x)dx - \frac{1}{N}\sum_{i=1}^{N} f(x_i) \Big| \leq \sum_{i=1}^{l} d_i \Big| \int_{[a^{(i)},b^{(i)}]} f(x)dx - \frac{1}{d_i N}\sum_{i=1}^{n_i} f(x_{ij}) \Big| \quad (3)$$

where $d_i = (\prod_{j=1}^{d}(b_j^{(i)} - a_j^{(i)}))^{-1}\frac{n_i}{N}$.

Combining Theorem 3, (2) and Lemma 3, we have

$$\sum_{i=1}^{L} d_i \Big| \int_{[a^{(i)},b^{(i)}]} f(x)dx - \frac{1}{d_i N}\sum_{j=1}^{n_i} f(x_{ij}) \Big| \leq \sum_{i=1}^{L} d_i \prod_{j=1}^{d}(b_j^{(i)} - a_j^{(i)})D_{n_i}^*(\tilde{X}^{(i)})\mathcal{V}^{[a^{(i)},b^{(i)}]}(f)$$

$$\leq \sum_{i=1}^{l} \frac{n_i}{N}\sqrt{\frac{N}{n_i d}}\frac{\theta}{c}D_{n_i,d}^*\mathcal{V}^{[a^{(i)},b^{(i)}]}(f)$$

$$\leq \sum_{i=1}^{l} \frac{n_i}{N}\sqrt{\frac{N}{n_i d}}\frac{\theta}{c}cd^{1/2}n_i^{-1/2}\mathcal{V}^{[a^{(i)},b^{(i)}]}(f)$$

$$= \frac{\theta}{\sqrt{N}}\sum_{i=1}^{L}\mathcal{V}^{[a^{(i)},b^{(i)}]}(f)$$

$$= \frac{\theta}{\sqrt{N}}\mathcal{V}^{[0,1]^d}(f)$$

where the last equality follows from Lemma 1. □

## 1.2 Proof for Corollary 5

*Proof.* In Monte Carlo methods, the convergence rate of $\frac{1}{n}\sum_{i=1}^{n} f(x_i)$ is of order $O(\frac{\text{std}(f)}{\sqrt{n}})$. Let $f(x) = \mathbf{I}\{x \in [a, b]\} = \mathbf{I}_{[a,b]}$ be defined on $[0, 1]^d$, we have $\text{var}(f) = P(A)(1 - P(A)) \leq 1/4$; thus, this error is bounded uniformly.

If another indicator function $\tilde{f}$ is defined on $[\tilde{a}, \tilde{b}] \subset (0, 1)^d$, then let

$$\phi_j(\tilde{x}_j) = \frac{a_j}{\tilde{a}_j}\tilde{x}_j\mathbf{I}_{[0,\tilde{a}_j)} + (a_j + \frac{b_j - a_j}{\tilde{b}_j - \tilde{a}_j}(\tilde{x}_j - \tilde{a}_j))\mathbf{I}_{[\tilde{a}_j,\tilde{b}_j)} + (b_j + \frac{1 - b_j}{1 - \tilde{b}_j}(\tilde{x}_j - \tilde{b}_j))\mathbf{I}_{[\tilde{b}_j,1]}$$

and $\phi(\tilde{x}) = \prod_{j=1}^{d}\phi_j(\tilde{x}_j)$ and apply Lemma 2, we have $\mathcal{V}^{[0,1]^d}(\tilde{f}) = \mathcal{V}^{[0,1]^d}(f)$; thus, the left term of (3) is bounded uniformly.

The error $| \int_{[0,1]^d} f_i(x)p(x)dx - \int_{[0,1]^d} f_i(x)\hat{p}(x)dx|$ is bounded by

$$\Big| \int_{[0,1]^d} f_i(x)p(x)dx - \frac{1}{n}\sum_{j=1}^{n} f_i(x_j) \Big| + \Big| \int_{[0,1]^d} f_i(x)\hat{p}(x)dx - \frac{1}{n}\sum_{j=1}^{n} f_i(x_j) \Big|$$

Now the result follows from triangular inequality. □

# 2 Experimental settings for comparison with KDE, OPT and BSP

## 2.1 Simulation environment

For each simulated dataset, we randomly generate the means of each component uniformly from the unit hypercube, and choosing $\sigma$ so that no two mixture components are closer than $3\sigma$ apart. We also

apply a transformation to each component to make it non-spherical, by multiplying the data by a random scaling and rotation matrix.

All methods under comparison are implemented in C++. The source codes for OPT and BSP were obtained from the authors. The code for KDE was obtained from: https://github.com/timnugent/kernel-density. We used the default parameters specified by the author of the software. The experiments were done on a MaxOS system with 16GB ram and a 2.2 GHz Intel Core i7 processor.

## 2.2 Error measure

The error measure used for comparison is the Hellinger distance, which can be used to quantify the similarity between two probability distributions. Let $P, Q$ be two probability measures with density functions $f$, $g$ respectively. Then the Hellinger Distance between $P, Q$ is defined as:

$$ H^2(P,Q) = \frac{1}{2} \int \left( \sqrt{f(x)} - \sqrt{g(x)} \right)^2 dx = 1 - \int \sqrt{f(x)g(x)} dx = 1 - \int \sqrt{\frac{f(x)}{g(x)}} g(x) dx $$

We estimate the distance between $p(x)$ and $\hat{p}(x)$ via importance sampling as follows:

$$ \hat{H}^2 = 1 - \frac{1}{n} \sum_{i=1}^{n} \sqrt{\frac{\hat{p}(x_i)}{p(x_i)}} $$

where $x_i \sim p(x)$. We take $n = 10^8$ through out our experiments.

## 2.3 Copula tranform

We found copula a helpful tool especially when the dimension is high. Specifically, we first estimate the marginal densities for each dimension $x_{\cdot j}$ ($j = 1, ..., d$), then make a copula transformation $z_{\cdot j} = \hat{F}_j(x_{\cdot j})$, where $\hat{F}_j$ is the estimated cdf of $x_{\cdot j}$ based on $x_{ij}, i = 1, ..., n$. We then estimate the joint density for the transformed variables. The reason for this is that for DSP (similar argument applies for OPT and BSP), the further the true distribution is from uniform, the more partitions are typically needed to capture the geometry of the density function. Copula can save the number of partitions by making the marginal distributions uniform within $[0, 1]$. We use copula transformation whenever the dimension is higher than 3. The maximum number of partitions for BSP, OPT and DSP are set to be 1000 for both the marginal distributions and joint distributions.

## 3 Two-phase DSP-kmeans

---
**Algorithm 1** Two-phase DSP-kmeans

---
**Input:** A dataset $X_N$ of size $N$
**Output:** $K$ initial centers for k-means
 1: **procedure** TWO-PHASE DSP-KMEANS($X_N, m, \theta, K$)
    Call DSP($X_N, m, \theta$) to get a partition with $L$ rectangles
 2:    **while** $L > K$ **do**
 3:       Look for the two sub-rectangles $r_i, r_j$ with lowest $d(r_i, r_j)$:

$$ d(r_i, r_j) = \min\{(\mu_i - \mu_j)^T S_i^{-1} (\mu_i - \mu_j), (\mu_i - \mu_j)^T S_j^{-1} (\mu_i - \mu_j)\} $$

       where $\mu_k, S_k$ are the sample mean and sample covariance matrix of points fall within $r_k$
 4:       Merge $r_i$ and $r_j$
 5:       $L = L - 1$
    Return the sample mean of the $K$ sub-rectangles

---

**Remark 3.1.** *When the number of points in a sub-rectangle is too large, one can restrict the covariance matrix to be diagonal to save computation.*

## References

[1] Art B Owen. Multidimensional variation for quasi-monte carlo. In *International Conference on Statistics in honour of Professor Kai-Tai Fang's 65th birthday*, pages 49–74, 2005.