[Reviews · NeurIPS 2016]

Reviewer 1

Summary

The authors propose a piecewise constant estimator of the density of a given sample in the unit hypercube. The estimation method is simple and easy to implement. It is based on a stepwise split of the hypercube into smaller hyperrectagles, in such a way that the data points are homogeneously spread within each subrectangle. The unknown density is then locally estimated by a uniform density on these rectangles.

Qualitative Assessment

The method is easy to implement, but it is not clear to me what is gained in comparison to local polynomial estimators. At a theoretical level, the estimator proposed in this work is weak, since the approximation bias (which is not studied at all in this paper) coming from approximating the true density by a piecewise constant function will be large as soon as the true density is not smooth enough.

Confidence in this Review

2-Confident (read it all; understood it all reasonably well)


Reviewer 2

Summary

The authors propose an adaptive partitioning method to fit a piecewise constant density estimate. They bound the integration error and get standard Monte Carlo bounds (O(1/\sqrt{n})) for their method. The method (or a variant---I am not quite sure) is tested on a set of synthetic data and a k-means initialization task.

Qualitative Assessment

Novelty: there are a fair number of adaptive partitioning density estimation methods in existence that the authors omit (Darbellay and Vajda, 1999; Petralia, Vogelstein, and Dunson, 2013; etc). Some of these are designed for high dimensions, some function in lower dimensions. In general, this space is fairly well studied but there is room for serious theoretical analysis. A much stronger result than the standard Monte Carlo integration rates would be something like rates in Hellinger or Lp distance, which are generally not dimension-free. These rates and associated traits (adaptivity to intrinsic dimension, shape constraints, etc) are generally much more predictive of how a method performs in the wild than simple MC integration rates. Impact: this method relies on star discrepancy to make partition selections and bound the error. However, this metric is generally impractical as it requires a supremum over all d-dimensional sub-rectangles, which is computationally burdensome. The authors suggest a work around, but this destroys their rate guarantees. This method would be quite slow in its original form and not theoretically supported in its modified form. Clarity: I had some trouble following this paper. Aside from some random typos, some of the ideas were not well explained. For instance, what is the intuition behind using gaps? How does the scaling step relate? Additionally, I found a few of the experimental numbers puzzling. KDE computations were quite slow, while computations with this method were quite fast. Given the general simplicity of a KDE and the complexity of star discrepancy, I would expect those numbers to be reversed. There must be some aspect of the implementation that is not well explained.

Confidence in this Review

2-Confident (read it all; understood it all reasonably well)


Reviewer 3

Summary

Interesting partitioning scheme. The improvement in speed is substantial. Although there is no gain in approximating power.

Qualitative Assessment

The results of the paper are rigorous and backed up by experiments. However the the approximations by simple functions is not novel. The partitioning scheme is a solid result.

Confidence in this Review

3-Expert (read the paper in detail, know the area, quite certain of my opinion)


Reviewer 4

Summary

This paper proposes an adaptive sequential partition method based on star discrepancy for density estimation. To make the proposed method adaptive to the input data, the authors introduce a novel statistic called "gap". The provable error bound and convergence rate are validated by the extensive experimental results including an interesting ML application of initialization of kmeans.

Qualitative Assessment

This paper proposes an adaptive sequential partition method based on star discrepancy for density estimation. To make the proposed method adaptive to the input data, the authors introduce a novel statistic called "gap". The provable error bound and convergence rate are validated by the extensive experimental results including an interesting ML application of initialization of kmeans. A significant speed-up on running time as shown in Table 2 is pretty impressive to me. I am not the expert in this field, but I find the motivation of the proposed method is very clear to me. In general, this paper is well executed. In Table 3, the authors report the improvement over random initialization of several well-known kmeans initialization methods. Although a discussion paragraph about the possible performance of other density estimators is given, I would like to see the experiments about whether the initialization step of other density estimators will contribute the majority time of entire kmeans procedure. In Line 20-21, the statement "kernel density estimation is ... higher than 6" is not obvious to me, please provide the corresponding reference or explain more.

Confidence in this Review

1-Less confident (might not have understood significant parts)


Reviewer 5

Summary

The paper proposes an efficient algorithm (providing average speedups of 100x) for non-parametric probability distribution estimation. The core concepts are very simple and easy to understand, yet the results are comparable to much more expensive and complex methods. At the end, the proposed algorithm is employed for choosing initial points for k-means clustering.

Qualitative Assessment

The paper is very clear and well written. I only have feedback on some minor issues: * In the introduction the partition is denoted as capital P = {\Omega^(1), ...}, while in the rest of the paper it's R = {r_1,...}. It'd be the best to keep this consistent. * In line 105 the last bin is said to be [a_j + (b_j - a+j)(m-2)/m, a_j + (b_j - a+j)(m-1)/m], but as far as I understand it should rather be [a_j + (b_j - a+j)(m-1)/m, a_j + (b_j - a+j)(m)/m] = [a_j + (b_j - a+j)(m-1)/m, b_j] (otherwise the last interval seems to be ignored). * X^{(i)} is used in the description of Algorithm 1, but it's only defined in line 128. * In line 2 of DSP procedure of Algorithm 1, I think that there's a typo - B should be R (the initial partition) - B isn't used anywhere else. * In line 10 of DSP procedure of Algorithm 1, Pr(r_i_1) = Pr(r_i) * |P(r_i_1)| / n_i, but P(r_i_1) hasn't been defined and as far as it seems to me it should rather be the number of data points that are contained in r_i_1 (it's not connected to Pr anyhow, so it doesn't seem to be a simple typo). Apart from the above remarks it was a pleasure to read the paper. The concept is very simple, yet achieves similar results to much more complex and computationally expensive methods.

Confidence in this Review

2-Confident (read it all; understood it all reasonably well)


Reviewer 6

Summary

This paper presents a discrepancy based sequential partition algorithm for estimating the density function. It exploits the concept of star discrepancy to model the uniformity of data. And authors derive some theoretical results to show that the proposed estimator is a good one with bounded approximation error and the convergence rate of the algorithm is as fast as Monte Carlo methods. The overall paper is solid and well written. However, my primary concern is that the computational complexity seems high as it involves computing multiple gaps of all dimensions for each recursively visited region. In the sense of practice, it may be less appealing.

Qualitative Assessment

Some notations are unclear. (1) In line 5 of algorithm 1, what does the superscript of a^{(i)} mean? (2) In line 10 of algorithm 1, what does |P(r_{i_1})| mean? It seems never been explained. In Algorithm 1, for each step of the ``for loop'', instead of computing gaps for all dimensions of each sub-region, could you randomly pick one or few such that the computational complexity is largely reduced for high dimensional problem? In section 5.1, several tricks are claimed to improve the efficiency of DSP. It is unclear that whether authors use all of them in the subsequent experiments or part of them. And one of them (replacing the star discrepancy with L2 star discrepancy) even makes the theoretical result does not hold any more. It would be great to compare the proposed algorithm with practically popular competitors like random forest. Also, it would be nice to see how the experimental results and convergence speed vary with different number of bins.

Confidence in this Review

2-Confident (read it all; understood it all reasonably well)